# Contextually Harmonious Local Video Editing

## Abstract

We introduce a new task for video editing: Contextual Harmonious Local Editing, which focuses on replacing a local moving subject in videos containing multiple subjects or reference objects. The goal is to ensure that the replaced subject maintains its original motion while its size remains harmonious with the scene's context. Previous methods often face two specific challenges when addressing this task: (1) ensuring the size of the replaced subject remains contextually harmonious (2) maintaining the original motion and achieving subject replacement without being affected by the motion of other subjects. To address the above problems, we propose a novel three stage video editing pipeline. We initially leverage large pre-trained models to acquire knowledge about the shape and size differences between the original and replaced subjects. To mitigate interference from context motion, we erase other moving subjects to extract the target subject's motion and dynamically choose the editing method to preserve the original subject's motion under different shape transformations. Following that, we seamlessly replace the original subject in the video with the resized edited subject, ensuring its size harmonizes with the video's context. As the first work to focus on this task, we also provide a high-quality evaluation dataset and metrics to assess the performance of existing methods on this task. Experimental results based on this dataset demonstrate that our method achieves state-of-the-art (SOTA) performance.

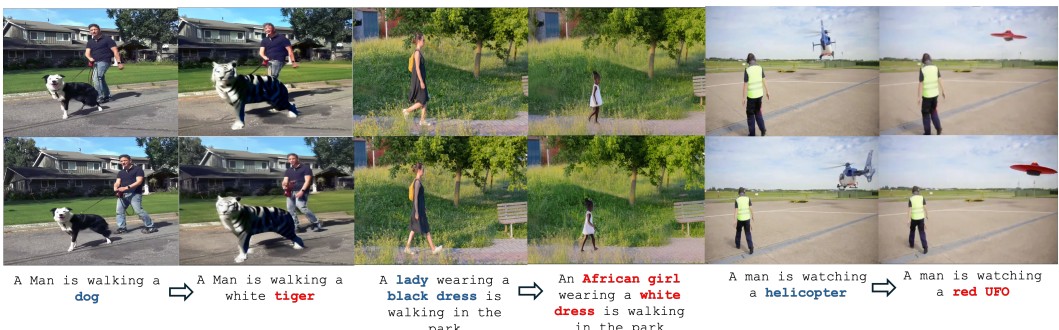

Figure 1: **Contextually Harmonious Local Video Editing.** The task aims to replace a single subject in videos containing multiple subjects or reference object, ensuring that the replaced subject maintains its original motion and appears harmonious in shape and size in relation to the others. (Left: dog → white tiger, enlarged relative to the other subject 'human'; Middle: lady → girl, shrinked relative to the reference object 'chair'; Right: helicopter → red UFO, drastic shape change.)

## 1 Introduction

In recent years, significant progress has been made in video editing Jeong et al. (2024b); Jeong & Ye (2023). However, in complex backgrounds or multi-subject scenes, the challenge of maintaining the motion of specific objects and adjusting their size to blend harmouniously with the scene remains unsolved. This research focuses on **contextually harmonious local video editing**, aiming to

replace specific subjects in multi-subject or reference-based backgrounds in a harmonious manner. This requires the replaced subject, under different shape transformations Qi et al. (2023), retains its original motion while adopting a size that harmoniously integrates into the video scene.

Accomplishing such editing involves addressing two key tasks: (1) To maintain harmony with other elements in the video, it's often necessary to adjust the subject's size when its semantics change. For instance, in Figure 1, when the "lady" transforms into a "little girl", her height relative to the bench should also change accordingly to maintain consistency in the scene. However, existing methods often struggle to understand how size should change with semantic shifts because they lack basic common sense. Additionally, it is challenging to scale the replaced subject according to a specific change ratio. To ensure continuity in the subject's movement during editing, it's crucial to map the original subject's feature points to the generated one and account for background changes due to the altered area. For example, when a "woman" is transformed into a "little girl", her shoulder and knee positions must be adjusted to align her actions with the scene, and the editing process should adjust the information of the background area she obscures. (2) The edited object should preserve the motion of the original subject. This challenge lies in the fact that the motion of the edited subject may be influenced by other subjects' motion in the video, as depicted in Figure 1(right), where two subjects exhibit significantly different motions—such as a person moving forward while a helicopter rotates—the model faces substantial challenges in accurately transferring specific motion characteristics between these divergent actions. This difficulty is rooted in the intrinsic complexities of capturing and interpreting distinct, simultaneous motion patterns within a unified framework. Moreover, this task becomes more complex due to varying requirements for motion preservation based on shape transformations, as seen in Figure 1, when the shape difference is small, fine-grained motions should be retained (dog → white tiger); however, with significant shape changes, only the dynamic patterns of the main keypoints need to be considered (helicopter → UFO).

To address these challenges, we propose a novel video editing pipeline that consists of three main stages: Motion and World Knowledge Acquisition, Shape Adapted Subject-centric Editing and Coarse-to-Fine Harmonious Subject Transfer. To help the model understand the knowledge regarding the extent of size changes before and after the subject transformation, we leverage large pretrained models to assess the relative size ratio between the original and replaced subject, aiding the edited subject in presenting a harmonious proportion within the video context. In addition, this module also perceives the moving objects in the video context and the shape differences of the edited subject relative to the original subject, providing prior knowledge for preserving the motion of the edited subject. In the Shape adapted subject-centric Editing stage, to prevent interference from the motion of other subjects during the editing process, we erase non-editing subjects and crop the target subject's motion region. Following this, we dynamically select the most suitable editing method based on the intensity of shape transformation between the original and target subjects, aiming at achieving motion preservation for the edited subject under different levels of shape variation. Finally, with the knowledge for size adjustment and the replaced subject, we introduce a Coarse-to-Fine Harmonious Subject Transfer stage to address the challenge of subject replacement and size transformation in multi-subject videos. In the Coarse Video Editing Phase, we obtain the motion of the resized edited subject, as well as the background information from the misalignment region between the edited subject and the original subject. Then, we achieve subject replacement based on these two parts with the original video which retains complete contextual information through our proposed intra-frame Spatial Guidance method. This achieves contextually harmonious replacement while preserving the non-edited areas. To maintain consistency in motion during the transfer of the resized subject, we further developed inter-frame temporal guidance, ensuring consistency between frames in the edited video and alleviate issues such as flickering and local distortion Jeong et al. (2024a) .

Lastly, as we are the first to focus on the issue of contextually harmonious local video editing, we have collected a dedicated video dataset that includes high-quality videos featuring various subject categories, encompassing scenes with multiple subjects and contextual reference objects. Additionally, we propose evaluation metrics to further assess the accuracy of local editing and the degree of background preservation in the edits.

**Main Contributions** (1) We introduce the task of "contextually harmonious local video editing" to address challenges in subject replacement and size adjustment within complex backgrounds or multi-subject scenes, ensuring subjects retain their original motion while blending harmoniously into the scene. (2) We develop an innovative video editing pipeline to effectively extracts and main-

tains the motion of target subjects while leveraging large pretrained models for world knowledge to guide seamless size adjustment and subject replacement. (3) We construct a high-quality video dataset featuring diverse subject categories and propose evaluation metrics to assess the accuracy and harmony of local edits. Our method achieves state-of-the-art performance on these metrics, demonstrating strong capabilities for contextually harmonious local video editing.

## 2 RELATED WORK

**Image Editing Models** For image editing methods, Prompt-to-prompt Hertz et al. (2022) and PNP Tumanyan et al. (2023) achieve the goal of editing images according to target prompts by manipulating attention features during the diffusion process. DragonDiffusion Mou et al. (2023) achieves drag-style image editing via gradient guidance produced by image feature correspondence Tang et al. (2023) in the pre-trained StableDiffusion Rombach et al. (2022) model and enables object replacement. Nevertheless, since it relies on image correspondence features Tang et al. (2023), it cannot handle significant size changes and maintaining the background depends on manually selected region mask information. Applying these image editing approaches to video frames directly will lead to serious issues such as flickering and inconsistency among frames. Our method inflates the T2I Diffusion Model Balaji et al. (2022) and constructs inter-frame Temporal Guidance, enabling to achieve better temporal consistency.

**Text Driven Video Editing** TokenFlow Geyer et al. (2023) and RAVE Kara et al. (2024) observe that features in diffusion models show similarities, offering fine-grained correspondences. FlowVid Liang et al. (2024), Fatezero Qi et al. (2023) and ControlVideo Zhang et al. (2023) use conditions such as depth maps and flow-warped videos to guide generation. However, their approach can only handle editing on objects with small shape changes because they indiscriminately utilize features extracted from the entire original video to guide the generation of the new video. Atlas Kasten et al. (2021) and CoDeF Lee et al. (2013) convert the video into a canonical image, perform editing on it, and then transform it back into a video. But When the video becomes complex, such as containing multiple moving subjects, this transformation becomes difficult to manage. Our method removes the motion of non-edited subjects, enabling it to handle videos with multi-subject scenes. By leveraging shape differences extracted from large language models, our method can effectively perform video editing with significant shape changes.

Existing methods are also focused on enabling precise multi-attribute editing. For instance, EVA Yang et al. (2024) enhances detailed local edits by mitigating attention leakage, strengthening attention scores for tokens associated with the same attribute. Nonetheless, if the subject of the edit undergoes significant size changes, the attention features representing different regions between frames cannot be calculated, thus making it impossible to handle size change issues. DynVideo-E Liu et al. (2024) is a character-centered video editing method that proposes the image-based video-NeRF editing pipeline. This approach primarily focuses on integrating the character into the video background but struggles to handle significant changes in the shape or size of the character. DreamMotin Jeong et al. (2024a) and Diffusion-Motion-Transfer Yatim et al. (2024) are capable of preserving the motion of the edited subject during the editing process. DMT extracts motion guidance by compressing the global features from the diffusion model. While this approach captures the motion patterns of the edited object, the loss of global features after compression makes it difficult to preserve the background in the edited video. Our method constructs guidance using intra-frame spatial consistency and inter-frame temporal coherence, enabling harmonious integration of subjects while preserving the overall video background.

## 3 METHODOLOGY

### 3.1 OVERVIEW

The Overview of our model is illustrated in Figure 2. Our method consists of three main satges: First, due to the impact of multiple subjects movements for local subject editing and the limitations of existing models in understanding shape and size differences between different subjects, we utilize a multi-modal large pretrained model Cheng et al. (2024) to capture contextual motion and employ a large language model Achiam et al. (2023) to gain knowledge about the relative changes in shape

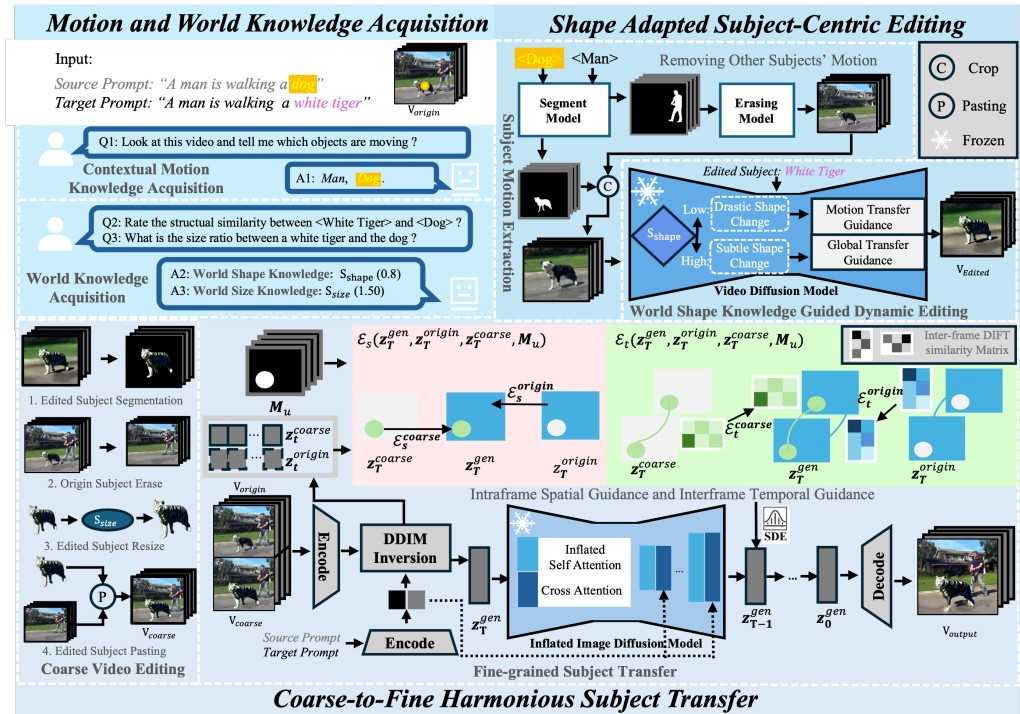

Figure 2: **Pipeline of our model.** The whole pipeline includes three key stages. Firstly, Large pre-trained model is adopted to analyze contextual motion in the video, as well as shape and size changes of the target subject relative to others. Then, the target subject's motion is extracted by removing non-target subjects and optimal editing method is dynamically selected for better edited subject in the Shape Adapted Subject-Centric Editing stage. Finally, in Coarse-to-Fine Harmonious Subject Transfer, we obtain the background information and the features of the resized subject needed for the subject transfer process during the coarse editing stage, and in the subsequent steps, we seamlessly accomplish subject replacement.

and size between the subjects, represented as $S_{shape}$ and $S_{size}$ in the Motion and World Knowledge Acquisition stage. Then, to prevent interference from the motion of other subjects during the editing of the target subject and to preserve the motion of the original subject under varying shape transformations, in the Shape Adapted Subject-Centric Editing stage, we erase non-target subjects Ravi et al. (2024); Zhang et al. (2022) and crop target subject's motion areas to isolate the target subject's motion. Based on $S_{shape}$, we dynamically choose suitable editing guidance for subject editing. Finally, to address the challenges of subject replacement with size transformation in multi-subject videos, we introduce the Coarse-to-Fine Harmonious Subject Transfer stage. We retain the motion of other subjects while erasing the original subject, then segment and resize the edited subject based on $S_{size}$, and subsequently paste it back into the video to obtain $V_{coarse}$. This process provides visual references for background and scaled subjects. In the subsequent Fine-grained Subject Transfer phase, we construct guidance using intra-frame spatial consistency and inter-frame temporal coherence to ensure the seamlessly subject replacement, maintaining overall fluidity and continuity.

## 3.2 MOTION AND WORLD KNOWLEDGE ACQUISITION

In order to acquire sufficient prior knowledge for achieving harmonious video editing, we propose leveraging a multi-modal large model and a large language model to capture motion and world knowledge as shown in the upper-left part of Figure 2 under different shape .

Initially, to disentangle the influence of different moving subjects, we utilize the ability of the vision-language models Cheng et al. (2024) to comprehend videos. By providing prompts "Look at this

video and tell me which objects are moving, respond only with object names" to the VLM, we can identify the moving subjects, which serves as preliminary support for isolating the independent motion of the editing subject. Then, considering that prior work has never addressed how size and shape difference affect motion preservation, we instruct the large language model Achiam et al. (2023) to apply its common sense in determining the similarity of the subject's shape before and after editing, along with calculating the ratio of the size change. Specifically, we design two prompts to enable the model to output $S_{\text{shape}}$ and $S_{\text{size}}$, which are defined as follows: $S_{\text{shape}}$ is obtained by asking, 'Not considering the size, from 0 to 1 please rate the structural similarity between source subject and target subject, only output a rate.' $S_{\text{size}}$ is derived from the prompt, 'Estimate the size ratio of an average source subject compared to an average target subject. Provide only a rate within the range of 0.3 to 3.' Through this approach, we determine the relative shape and size variations of subjects before and after replacement, providing a reference for selecting editing methods and ensuring the size of the replaced subjects appears harmonious within the video.

## 3.3 SHAPE ADAPTED SUBJECT-CENTRIC EDITING

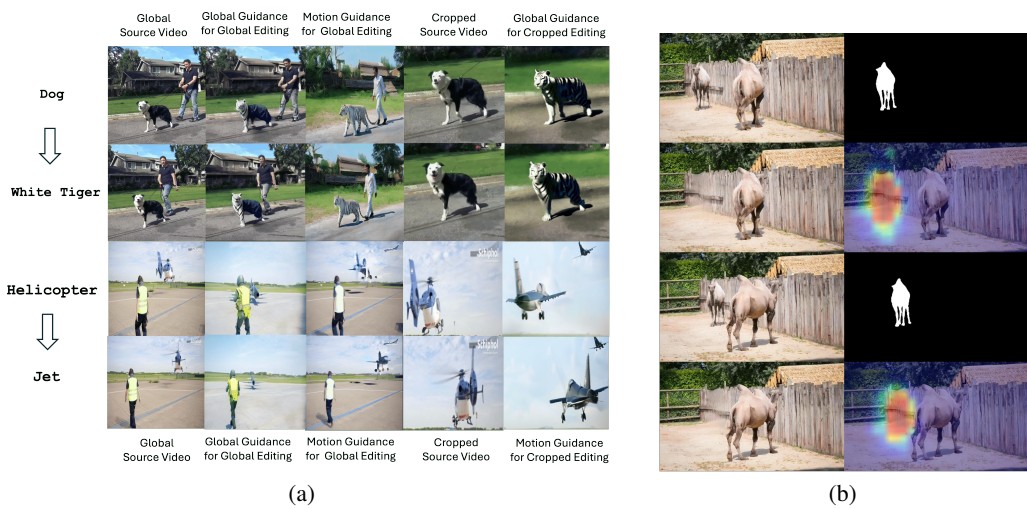

Figure 3: **Toy Experiments.** (a) Editing results of various guidance methods under different shape transformations and on videos with cropped regions. (b) Visualization of appearance changes between frames after the erasure operation using the difference matrix of LPIPS intermediate layers. This change is evaluated relative to the original video frames, red highlighting significant differences before and after processing.

In order to make the edited subject better preserve the motion of the original subject under different shape transformations scenarios. We designed the toy experiments as shown in Figure 3(a).

In our experiments, we applied Global Guidance and Motion Guidance following two existing SOTA methods Jeong et al. (2024a); Yatim et al. (2024) to edit local subject within a global video containing multiple subjects. However, as shown in column 2, the generated videos either failed to maintain the overall motion trajectory of the original subject (helicopter→jet), or failed to preserve the finer motion details effectively (dog→white tiger). We speculate the issue stems from interactions between different motions in the global video interfering with local subject editing. To verify this speculation, we extracted the motion of the target subject and edited it. We found that the replaced subject in the generated video demonstrated a better preservation of both overall and detailed motion. This indirectly suggests that the motion between different entities may interfere with each other, thereby affecting the local editing effects on specific subjects within the overall video.

Furthermore, we realize that existing editing approaches each have their advantages. As shown in column 3, Motion Guidance editing Yatim et al. (2024) excel in handling significant shape changes but struggle to preserve the original video's fine-grained motion with subjects of similar shapes. In contrast, Global Guidance editing Jeong et al. (2024a) can generate edited subjects while maintaining fine-grained motion, but is less effective with drastic shape changes.

Based on the observations mentioned above, we adopted Shape Adapted Subject-Centric Editing: We first utilize a segmentation model Ravi et al. (2024) to extract masks of moving objects within the video based on the previously obtained contextual motion information. Then, the original video and the masks of non-editable subjects are served as inputs for a video erasing method Zhang et al. (2022) to remove other moving subjects from the video, preventing them from affecting the motion extraction of the target subject. To further preserve the target's motion relative to the context in the original video and to better focus on the appearance replacement of the editing subject, we calculate the union space of the area occupied by the original target across frames in the video and cropped this portion for subsequent editing processes based on the mask of the original target. In order to select the more advantageous model under different circumstances, we dynamically adjust the model selection based on shape knowledge $S_{\text{shape}}$ acquired from the previous stage. When there is a drastic shape difference between the replacement subject and the original subject, we employ the Motion guidance method Yatim et al. (2024). Conversely, when the shape difference is subtle, we use the Global guidance method Jeong et al. (2024a). This design enables the model to achieve better motion preservation of the original subject during the replacement process, regardless of the degree of shape transformation.

### 3.4 COARSE-TO-FINE HARMONIOUS SUBJECT TRANSFER

The Coarse-to-Fine Harmonious Subject Transfer stage transfers the original subject to the edited subject while preserving the video's background, ensuring the edited subject remains contextually harmonious in size and motion. This stage is composed of two processes: Coarse Video Editing and Fine-grained Subject Transfer.

**Coarse Video Editing.** To ensure that the edited subject replaces the original subject in the video, maintaining harmonious size and motion, we proposed a Coarse Video Editing process. Using a strategy similar to the previous step, we first erase the original subject in the video using a video erasure model Zhang et al. (2022). Then, we resize the edited target subject based on $S_{\text{size}}$ and paste it into the position of the erased subject in the original video to create a coarse edited video $V_{\text{coarse}}$. The position for pasting is determined by aligning the bottom centers of the bounding boxes derived from the masks of both the original and edited subjects. Such editing process captures key information about the target subject's appearance and motion, as well as background details in areas where the shapes of the original and target subjects do not align, which is especially important when the edited subject is smaller than the original.

**Fine-grained Subject Transfer: Intra-frame Spatial Guidance.** During the coarse editing phase, thanks to our previous subject-centric editing design, a high-quality, motion-preserving, harmoniously resized edited subject can be presented in the coarse edited video, becoming the main target for subsequent subject transformation. However, as illustrated in Figure 3(b), the erasure process affects not only the corresponding area of the erased subject mask but also its surroundings. This suggests that erasure might impact nearby motion or background structures. To prevent this from affecting the final generated video, we plan to extract only the resized edited subject and the misaligned background area from the coarse edited video, while striving to keep the original scene's background unchanged. For this purpose, we designed the intra-frame Spatial Guidance, which is based on two functional components that correspond to our two objectives: First, to achieve motion retention of the resized generated subject and the misaligned background information from Coarse Edited Video, we designed $\mathcal{E}_S^{\text{coarse}}$ :

$$\mathcal{E}_S^{\text{coarse}} = \frac{1}{\alpha + \beta \cdot \mathcal{S}_S \left( \mathbf{F}_t^{\text{gen}}, \mathbf{F}_t^{\text{coarse}}, \mathbf{M}_{\text{u}} \right)} \tag{1}$$

Here, $M_u$ is obtained by calculating the union of the masks between the resized edited subject and the original subject. Based on this mask, the resized edited subject and the background information of misaligned region can be accurately extracted from the roughly edited video. In a similar way, to keep the background of the original video in the generated video, we designed $\mathcal{E}_S^{\text{origin}}$:

$$\mathcal{E}_S^{\text{origin}} = \frac{1}{\alpha + \beta \cdot \mathcal{S}_S \left( \mathbf{F}_t^{\text{gen}}, \mathbf{F}_t^{\text{origin}}, 1 - \mathbf{M}_{\text{u}} \right)} \tag{2}$$

where $\mathbf{F}_*$ is from the extracted intermediate features in the UNet decoder. Specifically, following Mou et al. (2023), Mou et al. (2024) , $\mathbf{F}_t^{\text{gen}}$ is extracted following from $\mathbf{z}_t^{\text{gen}}$ at the current time step, while $\mathbf{F}_t^{\text{origin}}$ and $\mathbf{F}_t^{\text{coarse}}$ come from $\mathbf{z}_t^{\text{origin}}$ and $\mathbf{z}_t^{\text{coarse}}$ stored in the DDIM Inversion process for $V_{origin}$ and $V_{coarse}$. and $\mathcal{S}_S$ is the energy function based on image feature correspondence to guide our model for subject transfer:

$$\mathcal{S}_S\left(\mathbf{F}_1, \mathbf{F}_2, \mathbf{mask}\right) = 0.5 \cdot \cos\left(\mathbf{F}_1[\mathbf{mask}], \text{sg}\left(\mathbf{F}_2[\mathbf{mask}]\right)\right) + 0.5 \tag{3}$$

Where $\text{sg}(\cdot)$ is the gradient clipping operation. In this way, the final conditional gradient Song et al. (2020) for intra-frame spatial guidance $\mathcal{E}_S$ is :

$$\nabla_{z_t}\mathcal{E}_S = \mathbf{M}_{\text{u}} \cdot \nabla_{z_t}\mathcal{E}_S^{\text{coarse}} + (1 - \mathbf{M}_{\text{u}}) \cdot \nabla_{z_t}\mathcal{E}_S^{\text{origin}} \tag{4}$$

**Fine-grained Subject Transfer: Inter-frame Temporal Guidance**   The above steps effectively achieve size-harmonious subject replacement while preserving the background, but they operate as a frame-independent optimization method without considering the temporal correlation between frames. Such per-frame operations can lead to localized distortions and notable flickering in the optimized frames Ma et al. (2023); Jeong et al. (2024a). To address this problem, we introduce inter-frame Temporal Guidance, which enhances feature consistency by leveraging the correlation of corresponding features between video frames. To construct inter-frame consistency, we similarly build an energy function between frames based on the correlation of intermediate layer features in the Unet according to Tang et al. (2023):

$$\mathcal{S}_T\left(\mathbf{F}, \mathbf{M}\right) = \{Sim\left(\mathbf{F}_i[\mathbf{M}_i], \mathbf{F}_j[\mathbf{M}_j]\right) \mid i, j = 1, 2, \ldots, n\} \tag{5}$$

Where $n$ is the total number of frames in the video, $i$ and $j$ correspond to the indices of different frames. For $X \in \mathbb{R}^{m \times c}$ and $Y \in \mathbb{R}^{n \times c}$, $Sim(X, Y)$ is the matrix obtained by calculating the cosine similarity between the corresponding row vectors $X[i, :]$ and $Y[j, :]$, with a size of $\mathbb{R}^{m \times n}$. $\mathcal{S}\left(\mathbf{F}, \mathbf{M}\right)$ is a set of inter-frame similarity matrices, consisting of $n^2$ matrices.

Moreover, based on the concept of constructing intra-frame Spatial Guidance, for background regions, we ensure the feature similarity matrix between frames is consistent with the original video, while for editing regions, it aligns with the coarse edited video. This is because subject motion from subject-centric editing is well preserved, but background inpainting distortions can accumulate over frames Zhou et al. (2023). Therefore, our guidance function based on the original video and the coarse-edited video is as follows:

$$\mathcal{E}_T^{\text{coarse}} = \frac{1}{\alpha + \beta \cdot \text{Mean}(\cos(\mathcal{S}_T\left(\mathbf{F}_t^{\text{gen}}, \mathbf{M}_{\text{u}}\right), \mathcal{S}_T\left(\text{sg}\left(\mathbf{F}_t^{\text{coarse}}\right), \mathbf{M}_{\text{u}}\right)))} \tag{6}$$

$$\mathcal{E}_T^{\text{origin}} = \frac{1}{\alpha + \beta \cdot \text{Mean}(\cos(\mathcal{S}_T\left(\mathbf{F}_t^{\text{gen}}, 1 - \mathbf{M}_{\text{u}}\right), \mathcal{S}_T\left(\text{sg}\left(\mathbf{F}_t^{\text{origin}}\right), 1 - \mathbf{M}_{\text{u}}\right)))} \tag{7}$$

Finally, the conditional gradient Song et al. (2020) for inter-frame temporal consistency $\mathcal{E}_T$:

$$\nabla_{z_t}\mathcal{E}_T = \mathbf{M}_{\text{u}} * \nabla_{z_t}\mathcal{E}_T^{\text{coarse}} + (1 - \mathbf{M}_{\text{u}}) * \nabla_{z_t}\mathcal{E}_T^{\text{origin}} \tag{8}$$

Total gradient is calculated as shown below, where $\mu_S$ and $\mu_T$ are weights for different guidance.

$$\nabla_{z_t}\mathcal{E} = \mu_S \nabla_{z_t}\mathcal{E}_S + \mu_T \nabla_{z_t}\mathcal{E}_T \tag{9}$$

**Fine-grained Subject Transfer: Inflating T2I Diffusion Model**   Since our inter-frame temporal guidance requires the model to enable inter-frame information interaction, we inflate this part following Wu et al. (2023a): modifying the U-Net's $3 \times 3$ convolutions to $1 \times 3 \times 3$ and replacing spatial attention with sparse spatio-temporal attention This uses frame embeddings as queries and current and adjacent frame embeddings as keys and values, ensuring better spatiotemporal context integration.

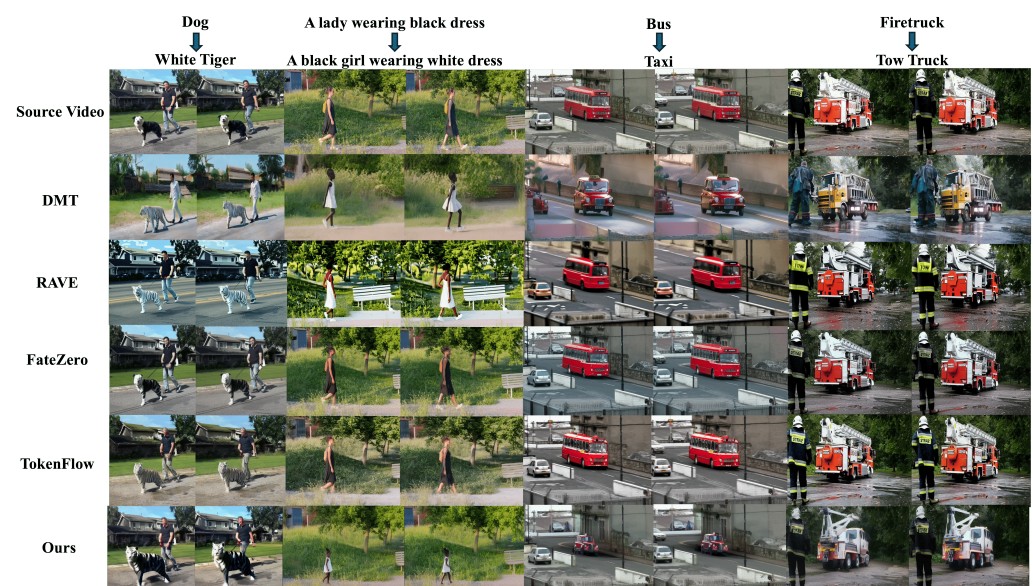

Figure 4: **Qualitative Results.** Our model can achieve local editing under various shape transformations, and present them at a harmonious size in the preserved video context.

## 4 EXPERIMENTS

### 4.1 EXPERIMENTAL SETTINGS

**Dataset** To build a multi-subject video dataset, we collected 40 videos from TGVE Wu et al. (2023b), DAVIS Caelles et al. (2019), YouTube-VOS Xu et al. (2018), Animal Kingdom Ng et al. (2022) and the Internet. Almost all of the raw videos have a resolution exceeding 512x512, with the majority surpassing 1920x1080. These raw videos vary in duration and are characterized by clarity, stationary camera motion, and the presence of other moving subjects or reference elements(such as furniture, trees) besides the main edited subject. To showcase the diversity of subjects in natural scenes, our edited video subjects include various categories, such as humans, animals, and vehicles. For each video, we select segments without subject occlusions to extract frames, resulting in video versions with 24 frames and 8 frames, allowing them to accommodate methods which need different editing lengths. Subsequently, we use the captions extracted by the VideoLLaMA2 Cheng et al. (2024) as the original text prompts, and then employ GPT-4 Achiam et al. (2023) to generate editing prompts aimed at replacing the original subject with a new subject, followed by manual adjustments. It is worth noting that, considering our task settings, we will generate editing prompts where both the shape and size of the original subject undergo some changes. Through the aforementioned process, we obtained a multi-subject video dataset consisting of 60 triplets, each containing two different-length videos (24 frames and 8 frames), the original prompt, and the editing prompt, which provides a sufficient quantity for the field of video editing.

### 4.2 QUALITATIVE RESULTS

In our experiments, we conducted three types of video editing which include editing objects into ones that may be similar in shape but resized to be smaller or larger, as well as into objects with significant shape changes. We compare our method with the following open-sourced State-of-the-Art video editing approaches: Diffusion motion transfer (DMT) Yatim et al. (2024), Fatezero Qi et al. (2023), Tokenflow Geyer et al. (2023) and Rave Kara et al. (2024). Qualitative results are shown in the figure 4. As observed in the fourth and fifth rows, FateZero and TokenFlow either fail to manage significant shape alterations or only perform texture replacement without modifying the original object's geometry. DMT is able to successfully edit the subject while meeting the requirements for shape and size changes but it fails to maintain consistency with the background

| | Automatic Metrics | | | | | Human Study | | | |
| --- | --- | --- | --- | --- | --- | --- | --- | --- | --- |
| | CLIP-T ↑ | CLIP-F ↑ | Dino ↑ | Masked LPIPS(target) ↑ | Masked LPIPS(background) ↓ | Edit-Acc ↑ | Frame-Con ↑ | BM-Preserve ↑ | Harmony-Score ↑ |
| FateZero | 29.90 | **96.78** | 64.73 | 0.01742 | 0.2052 | 2.31 | 3.95 | 3.95 | 2.74 |
| Rave | 31.20 | 95.71 | 66.53 | 0.03030 | 0.3607 | 2.93 | 3.76 | 3.22 | 2.37 |
| Tokenflow | 31.01 | 96.32 | 65.43 | 0.01774 | 0.2562 | 2.42 | 3.77 | 3.53 | 2.25 |
| Dmt | 31.43 | 95.90 | 71.96 | 0.04559 | 0.5102 | 3.64 | **4.05** | 2.62 | 3.18 |
| Ours | **31.54** | 95.36 | **73.20** | **0.04927** | **0.2083** | **4.26** | 3.85 | **3.90** | **4.05** |

Table 1: **Quantitative evaluations.**

of the original video. For instance, in the left video of the second row, DMT also changes the original person's clothing to white. RAVE can achieve some required object changes specified in the text, and it maintains the background quite well. However, if the differences in size and shape between the edited object and the original object are large, it can not handle well. For example, after transforming the bus into a taxi, the size and shape structure of the subject remain unchanged.

Thanks to the incorporation of world knowledge, the selection of editing methods based on the degree of shape changes, and the design that maintains consistency after size changes in both spatial and temporal dimensions, our approach robustly edits both the size and shape of objects while preserving the background structure and appearance of the original video, effectively addressing the issues present in the results of other methods. More results shall be viewed here.

### 4.3 QUANTITATIVE COMPARISON FOR MULTI-OBJECT LOCAL EDITING

We conducted a quantitative evaluation on our proposed dataset. In addition to following previous works by using CLIP-T and CLIP-F Radford et al. (2021) to assess video-text consistency and inter-frame consistency, we introduced the Dino Liu et al. (2023) to more determine whether the subject replacement was successfully edited, as previous works Yuksekgonul et al. (2023) have shown that CLIP struggles to understand combination information in multi-subject video captions. (We ensured that the replaced subject does not exist in the original video.) To evaluate the preservation of non-edited areas, we combined a segmentation model with LPIPS distance Zhang et al. (2018) (see Table 1, Masked LPIPS (background)). We segmented the replaced subject in all generated videos based on its text description; if the segmentation model failed, we used the original subject's mask instead. We calculated the union of the masks for each edited subject and used it to mask areas when computing the LPIPS score between edited and original videos, ensuring a fair comparison of non-edited area preservation. Similarly, we assessed the similarity between the edited and original videos in the original subject's area (see Table 1, Masked LPIPS (target)) by calculating the LPIPS score within the corresponding mask. Given that other models often fail to edit effectively under our setup (see Figure 4), this metric indicates whether the original subject has been altered, with a lower score signifying unsuccessful replacement. As shown, our method achieved the best results in Dino, CLIP-T, and Masked LPIPS (target), and also obtained the best scores in Masked LPIPS (background), indicating that our method successfully performs editing while preserving the background consistency of the original video as much as possible. Although our method scores slightly lower in CLIP-F, it may be because other methods, which have a lower success rate of editing on our dataset, generate videos that are more similar to the original, thus preserving the original video's motion Cong et al. (2023).

*User study.* 25 Participants are involved and asked to watch the input video and then the anonymized output videos from each baseline. They were then asked to rate four questions on a scale of 1 to 5: (i) Edit Accuracy (Edit-Acc): Does the output video accurately reflect the target text by appropriately editing all relevant elements? (ii) Frame Consistency (Frame-Con): Are the frames in the output video temporally consistent? (iii) Background and Motion Preservation (BM-Preseve): Have the motion and non-edited parts of the input video been accurately preserved? (iv) The harmony of size change (Harmony-Score): Is the size change reasonable in the output video? Table 1 shows that our method demonstrates excellent performance across all evaluation metrics, performing well in inter-frame consistency and successfully maintaining a reasonable size for the edited subject in the original video.

### 4.4 ABLATION STUDY

In view of the relative independence of the various modules in our method, the ablation here mainly focuses on the Fine-grained Subject Transfer module. As shown in the process of transferring a dog

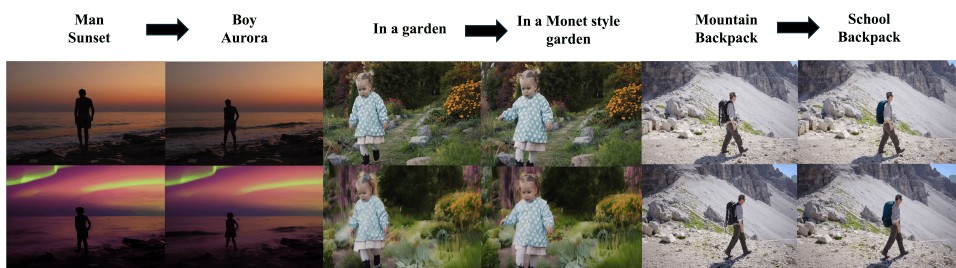

Figure 5: **Ablation Study for Fine-grained Subject Transfer.** With the introduction of various modules, the integration between the subject and the scene becomes increasingly seamless.

Figure 6: **General Editing.** Our method can be adapted to handle common video editing tasks like background replacement(left), style transformation (middle) and part modification (right).

to a white tiger in Figure 5, directly copy-pasting introduces issues such as the loss of contextual details like shadows and leashes, as well as jagged and unnatural edges around the subject. This leads to an integration that appears disjointed and lacks harmony. By utilizing Intra-frame Spatial Guidance technology, we can seamlessly blend the edited subject with the original video background, and reintroduce background details during the editing process. However, since this process is essentially a single-frame editing operation, it is prone to issues such as distortion and flickering. By utilizing the inflated image diffusion model and introducing Inter-frame Temporal Guidance, we leverage inter-frame constraints to mitigate the impact of such issues.

## 4.5 GENERAL LOCAL EDITING ABILITY

As shown in Figure 6, our method can be adapted to handle common video editing tasks previously addressed by other works (e.g., TGVE Wu et al. (2023b)), such as background replacement, style transformation and part modification. Our method offers significant flexibility for editing by merging attributes from multiple edits. For example, we can adjust the subject's size harmonious with background changes, alter the background style while preserving the original subject and change only the backpack of a person.

## 5 CONCLUSION

Our proposed video editing pipeline addresses the task of contextually harmonious local video editing. This approach tackles the challenges of subject replacement and size adjustment in complex backgrounds and multi-subject scenes, ensuring that edited subjects retain their original motion characteristics and fit harmoniously into the video context at an appropriate size and shape. To validate our approach, we created a specialized video dataset focused on contextually harmonious local video editing. This dataset comprises high-quality videos featuring a wide range of subject categories, including scenes with multiple subjects and contextual reference objects. Experimental results demonstrate that our method achieves state-of-the-art performance in such setting.

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
