# OpenReview forum: "Contextually Harmonious Local Video Editing"
_ICLR.cc/2025/Conference — ICLR 2025 Conference Withdrawn Submission_

### Official Review · Reviewer_A3XB · 2024-10-27

**Soundness:** 2
**Presentation:** 2
**Contribution:** 1
**Rating:** 3
**Confidence:** 5

**Summary:**

This paper proposes specific condition for the video editing as "contextually harmonious local video editing", where the proposed method of erasing moving subjects to extract the target subject's motion. This paper claims that it dynamically choose the editing method to preserve the original subject's motion under different shape transformation.

**Strengths:**

[1] Figure is illustrative.

[2] Experiments are sufficient.

[3] Writing is easy to follow.

**Weaknesses:**

[1] Contextually harmonious video editing is not a new concept. Traditional video editing always strives for harmonious integration without altering unintended attributes. Can the authors clarify the distinction between standard video editing and the proposed "contextually harmonious video editing"?

[2] Consistency and preservation of attributes seem lower than recent standards in video editing quality. For instance, the man's face in Figure 1 appears noticeably altered.

[3] Are there video results available? I can not validate the consistency by only two resulting frames in the paper. The sample provided appears to exhibit noticeable flickering, which may disrupt visual continuity.

[4] How do the authors measure harmony? The concept of harmony is highly abstract and subjective. Is there an objective metric used in this study?

**Questions:**

Please see the weakness. My questions are based on weakness.

---

### Official Review · Reviewer_a5ch · 2024-11-01

**Soundness:** 2
**Presentation:** 2
**Contribution:** 1
**Rating:** 3
**Confidence:** 4

**Summary:**

This paper presents the method of local editing, which focuses on replacing a local moving subjects in videos containing multiple subjects or reference objects. To that end, it utilizes pre-trained VLM model to extract world knowledge on source and target object and modify the video editing pipeline to achieve an alignment between edited and non-edited areas.

**Strengths:**

Belows are strong points that this paper have:

1. This paper aims to divide-and-conquer by realizing the limitations and proposing reasonable techniques to solve them. (e.g., utilize world model to capture the relationship between source and target instance, design the pipeline to achieve the misalignment between edited and non-edited area).

2. Fair presentation to make overall paper to be easy to read.

**Weaknesses:**

Here are some weaknesses in this paper:

1. Although the paper aims to implement a divide-and-conquer approach, it heavily relies on pre-existing models or methods to enhance its pipeline, which raises questions about its novelty as a research contribution. For instance, the pipeline depends significantly on preliminary support from Vision-Language Models (VLMs) for extracting moving objects and gathering additional information, such as size. This dependency introduces potential failure points: what if the VLM model fails to detect the objects accurately or incorrectly estimates the size ratio between the source and target objects? Such reliance could lead to various failure cases stemming from the external model’s limitations.

2. The contribution to temporal consistency is questionable. The authors employ feature consistency based on an inter-frame DIFT similarity matrix, but it remains unclear if this approach can fully address pixel-level flickering issues. Aligning with this concern, Table 1 shows that the proposed method underperforms in terms of temporal consistency, highlighting a critical weakness.

3. The qualitative evaluation is limited without accompanying videos. Providing some qualitative video examples would enhance the presentation of effectiveness.

4. Any reason of choosing 40 videos out of 76 TGVE dataset?

**Questions:**

Please refer to above Weaknesses.

---

### Official Review · Reviewer_uatC · 2024-11-02

**Soundness:** 2
**Presentation:** 2
**Contribution:** 2
**Rating:** 3
**Confidence:** 5

**Summary:**

The authors introduce a novel method to address the local editing task of replacing subjects in multi-subject videos based on text guidance. Their approach is specifically designed to manage size discrepancies and maintain motion consistency throughout the editing process. To achieve this, they propose a pipeline comprising three stages: utilizing a large language model to gather information on the size and shape of the target object, performing the initial editing, and refining the results through a coarse-to-fine approach.

**Strengths:**

- The paper leverages a large multi-modal pretrained model to determine shape and size ratios, providing a feasible and easily applicable solution. This is a clever approach, effectively addressing the size discrepancy issue in local video editing.

- The method offers a simple yet effective solution to address size discrepancies by aligning bottom centers during the coarse video editing stage. However, this alignment is best suited for objects grounded on a flat surface (e.g., a walking person or a moving car). For objects in mid-air or without a clear floor reference, this approach may inaccurately determine the correct position. It would be beneficial for the authors to clarify this limitation, perhaps by noting that their videos primarily feature grounded objects.

- Visual comparisons provided via the Google Sites link are valuable and showcase the results especially when big size changes are needed.

- The user study on harmony score demonstrates the method's improvement on claimed problem over previous approaches, particularly in terms of accurately handling size effects in generated outputs.

**Weaknesses:**

- The paper claims to be the first to focus on local video editing (line 100), a statement that is questionable given the existence of recent works like "AVID: Any-Length Video Inpainting with Diffusion Model" (CVPR 2024), "Videoshop: Localized Semantic Video Editing with Noise-Extrapolated Diffusion Inversion" (ECCV 2024), and "VideoSwap: Customized Video Subject Swapping with Interactive Semantic Point Correspondence" (CVPR 2024). While VideoSwap requires a reference image, this requirement does not diminish its relevance to local video editing. Given these prior works, it would be beneficial for the authors to clarify the distinctions of their method. Specifically, I would like to understand the differences between applying one of these methods with a provided object mask versus using the proposed method (which also extracts the object mask using a segmentation model). A thorough discussion on these distinctions would enhance the clarity and novelty of the paper.

- The paper suffers from significant writing issues that require major revisions. The language is often unclear, and there are numerous grammatical errors, which hinder the readability and impact of the work. Additionally, Figure 2 is particularly difficult to follow in its current form. Splitting this figure into three distinct modules or sections could make each stage more comprehensible. Alternatively, restructuring Figure 2 with improved labels and visual cues could help readers better understand the pipeline, especially when cross-referencing it with the manuscript.

- The paper relies on Vision-Language Models (VLMs) to assess structural similarity and size ratios, yet it lacks a detailed discussion on the reliability of these models in less common scenarios. For instance, it is unclear how the model performs when tasked with identifying rare or unusual objects. Experimental results or evidence demonstrating the VLM's effectiveness and limitations in such cases would greatly strengthen the paper's claims.

-  The paper does not provide any information on runtime or GPU requirements, which are critical for assessing the practical feasibility of the proposed method. Including a comparative analysis in terms of computational cost and runtime would offer valuable insight into the method's scalability and reliability, particularly for real-world applications where these factors are crucial.

**Questions:**

Minor:

- Line 69: A space is needed after "Figure 1" (currently written as "Figure 1(right),").
- Line 146: There is a typo; "DreamMotion" should be corrected.
- Line 147: The abbreviation "DMT" is used without prior definition. It likely stands for "Diffusion-Motion-Transfer," but it should first appear - in full with the abbreviation in parentheses.
- Line 158: There is a typo; "stages" should be corrected.
- There are many other typos with paranthesis where a space is required before the paranthesis.
- In the table presenting quantitative results, specifically the masked LPIPS metric and the BM-preserve metric from the user study, FateZero has a better score than the proposed method; however, this score is not bolded. This inconsistency should be corrected to accurately show the results.

---

### Official Review · Reviewer_ahCd · 2024-11-03

**Soundness:** 3
**Presentation:** 3
**Contribution:** 3
**Rating:** 6
**Confidence:** 4

**Summary:**

The proposed method targets at the contextual harmonious local video editing, which is a new task for video editing. The method is three-stages. It firstly uses the pretrained LLM to estimate the size of edited subject in the scene's context. Then, it erases non-editing subjects and focus on the target subject motion. Finally, it introduces the coarse-to-fine subject transfer for final video editing. Experiments demonstrate the superiority of the proposed method.

**Strengths:**

++ This paper is interesting in researching on the size of edited subject to make it harmonious with original video.

++ Using pretrained models to get world knowledge of subject size is reasonable.

++ The proposed method achieves much better performance than previous methods.

**Weaknesses:**

-- Although the paper achieves much better performance, its three-stage pipeline (final stage is a coarse-to-fine stage) is a bit tedious and might not be easy to use.

-- The paper requires pre-estimated masks using pre-trained models. However, the estimated masks could be inaccurate. Therefore, it is necessary to evaluate the method's robustness with respect to the accuracy of masks.

-- No edited videos are provided. It is necessary to evaluate the method's temporal performance.

**Questions:**

Q1: Could the method support multi-subject editing?

Q2: What might be the failure cases for the method?

---

### Note · Authors · 2024-11-15

I have read and agree with the venue's withdrawal policy on behalf of myself and my co-authors.